# Dietary Habits of a Group of Children with Crohn’s Disease Compared to Healthy Subjects: Assessment of Risk of Nutritional Deficiencies through a Bromatological Analysis

**DOI:** 10.3390/nu14030499

**Published:** 2022-01-24

**Authors:** Flavio Labriola, Caterina Marcato, Chiara Zarbo, Ludovica Betti, Arianna Catelli, Maria Chiara Valerii, Enzo Spisni, Patrizia Alvisi

**Affiliations:** 1Pediatric Gastroenterology Unit, Maggiore Hospital, Largo Bartolo Nigrisoli, 2, 40133 Bologna, Italy; caterina.marcato@ausl.bologna.it (C.M.); chiara.zarbo@ausl.bologna.it (C.Z.); patrizia.alvisi@ausl.bologna.it (P.A.); 2Specialty School of Pediatrics—Alma Mater Studiorum, Università di Bologna, 40138 Bologna, Italy; ludovica.betti@studio.unibo.it (L.B.); arianna.catelli@studio.unibo.it (A.C.); 3Department of Biological, Geological and Environmental Sciences, University of Bologna, Via Selmi 3, 40126 Bologna, Italy; chiaravalerii@hotmail.it (M.C.V.); enzo.spisni@unibo.it (E.S.)

**Keywords:** Crohn’s disease, dietary intake, nutritional deficiencies, micronutrients, bromatology

## Abstract

Diet is a matter of interest in the pathogenesis and management of Crohn’s Disease (CD). Little is known about CD children’s dietary habits. Our aim was assessing the quality and the amount of nutrient intake in a group of CD pediatric patients. Data were compared with those of healthy subjects (HS). In total, 20 patients (13 males) and 48 HS (24 males) aged 4–18 years were provided with a food diary to fill out for one week. Winfood software performed the bromatological analysis, providing data about intakes of proteins and amino acids, fatty acids, carbohydrates, cholesterol, fibers, minerals, vitamins, and polyphenols. Estimates of the antioxidant activity of foods and of the dietetic protein load were also calculated. The diet of CD patients was poorer in fibers, polyphenols, vitamin A, beta-carotene, and fatty acids, and richer in animal proteins, vitamin B12, and niacin. PRAL was higher in CD patients’ diets, while ORAC was higher in HS. No significant differences were observed in carbohydrate and other macro- and micronutrient consumptions. CD dietary habits seem to reflect the so-called Western diet, possibly involved in CD pathogenesis. Furthermore, analysis of dietary habits allows for prevention of nutritional deficiencies and timely correction through education and supplementation.

## 1. Introduction

Crohn’s disease (CD) is an Inflammatory Bowel Disease (IBD) characterized by segmental inflammation involving any part of the gastrointestinal tract from mouth to anus [1]. Approximately 20% of CD cases are diagnosed before 18 years of age [2]. The burden of CD is high in most Western countries and seems to be rapidly increasing in developing countries [3,4,5].

Diet is a key environmental factor in the pathogenesis of CD, likely acting as a pro-inflammatory agent [6]. In the developed “Western” countries, diet has undergone several changes from the traditional diets of the past when the prevalence of CD was lower. Mainly, alimentary habits switched from plant-based to animal-sourced foods, but other major changes, such as increased dairy fats and the adoption of refined wheat, emulsifiers, and thickeners, could be associated with intestinal inflammation, as demonstrated using animal models [6,7].

On the other hand, many adult IBD patients perceive diet to be a worsening factor on their own health status, and food restriction is commonly considered as a useful tool to modify disease activity, thus trying to prevent acute manifestations. Consequently, diet changes derive from self-experience rather than being supervised by a nutrition professional, and this increases the risk of nutrient deficiencies [8]. 

Dietary habits in children and adolescents with CD have been investigated in a few studies [9,10,11,12], but conclusions are difficult to draw because of methodological differences and limitations, such as the retrospective assessment of the diet. 

The aim of the present study is to assess the dietary intakes of a group of children suffering from CD, in comparison with similar Healthy Subjects (HS) by age. This observational study aimed to analyze the quality of the diet and the amount of macro- and micronutrient intake. 

## 2. Materials and Methods

Sixty-eight subjects aged more than 4 years and less than 18 years were enrolled from September 2019 to September 2020 at the Pediatric Gastroenterology Centre (Maggiore Hospital, Bologna, Italy). 

Among them, 20 were patients diagnosed with CD, while 48 were healthy subjects.

Diagnosis of CD was made according to the Revised Porto Criteria [13].

Patients with complicated CD (e.g., imminent need for surgery for strictures or fistulizing disease) were excluded. To reduce bias, we also excluded patients whose parents reported changes in their diet shortly before or after the diagnosis.

Clinical features of patients, such as disease activity and location, were recorded. According to the Paris Classification [14], L1 disease is ileocecal CD, L2 stands from Crohn’s colitis, L3 involves both ileum and colon, while in L4 disease there is an upper gastrointestinal involvement. Perianal involvement may or may not be associated with one of the disease locations. The Pediatric CD Activity Index was used to assess disease remission (<10) or activity (>10) [15]. 

Written informed consent was obtained from each participant’s parents and detailed information about the study was provided. The study was approved by the Ethics Committee (Prot. N. 11116828 of 29 May 2019). 

### 2.1. Study Design

After enrollment, a weekly food diary was provided to participants to be anonymously filled out. Families received instructions on how to record food intakes, including quantities and qualities of foods and dressings. The contribution of oral supplements was not considered, so that only food-derived nutrients were counted. All the enrolled subjects were asked to maintain their normal dietary habits. After filling out their diaries, participants returned them to the reference center. Records were entered in a specific software (Winfood, Medimatica Srl Unipersonale, Colonnella (TE), Italia) used to perform bromatological analysis. This provides estimates of macro- and micronutrient contents based on the Italian food composition database described by Sette et al. [16]. Data obtained were thus exported in an Excel database. For each patient or healthy subject, 7-day median of each nutrient intake was calculated.

Nutrients considered for bromatological analysis were proteins (total, animal, and vegetable), amino acids, saturated and unsaturated fatty acids, carbohydrates (available carbohydrates, amid, oligosaccharides), cholesterol, total, soluble, and insoluble fibers, minerals and trace elements (calcium, iron, sodium, potassium, chloride, chromium, iodine, fluorine, magnesium, copper, selenium), vitamins (A, B1, B2, B6, B9, B12, C, D, E), and polyphenols. Total calories were also counted. Furthermore, through Winfood software analysis, we were able to obtain data about ORAC (Oxygen Radical Absorbance Capacity) and PRAL (Potential Renal Acid Load). ORAC is an index that estimates the in vitro antioxidant food activities. PRAL is an indirect measure of the biochemical balance of acidifying and alkalizing molecules contained in all foods. A positive score indicates the presence of acidifying potential renal load (typically meat and dairy products), while a negative score is obtained by a large assumption of alkalinizing foods such as vegetables and fruits. 

### 2.2. Statistical Analysis 

Continuous variables are expressed as mean ± Standard Deviation (SD) or median and Interquartile Range (IQR), accordingly. The Kolmogorov–Smirnov assessed normality of distributions. The Mann-Whitney test was performed to compare medians of 7-day nutrient intakes among patients and healthy subjects. Comparisons of other means or medians were made with Student’s t-test or Mann-Whitney test, when appropriate. Chi-square test was used to assess association between categorical variables. A *p* value of <0.05 was considered statistically significant. Analyses were performed using IBM SPSS statistic version 28 (IBM SPSS Statistics, Armonk, NY, USA: IBM Corporation).

## 3. Results

This was a cross-sectional observational study conducted on 20 pediatric patients diagnosed with CD and 48 HS. 

The CD group was made up of 13 boys and 7 girls, and the mean age was 15 years (range 9–18, Standard Deviation [SD] ± 2.53), whereas the HS group was made up of 24 boys and 24 girls, and the mean age was 11 years (range 6–18, SD ± 4.87). The two samples did not significantly differ in terms of sex distribution (*p* = 0.2957). All enrolled subjects were Caucasian and of Italian nationality, thus minimizing the diet variability due to different food and cultural traditions. 

In the CD group, 17 children were in clinical remission (average PCDAI 1.1, SD ± 2.2), while three showed mild disease activity (average PCDAI 21.7, SD ± 5.8). The most frequent disease location was L3 (*n* = 15), while three patients had L1, one had L2, and one had perianal disease. 

The total energy daily intake showed no significant differences among CD (1440.04 ± 341.63 Kcal/day) and HS (1400.5 ± 355.69 Kcal/day) groups (*p* = 0.43). 

Overall, we observed some significant differences between the two groups both for macronutrient and micronutrient intake.

The most relevant significant results are shown in Figure 1.

### 3.1. Fibers

The daily ingestion of fibers was lower in CD than in HS. The difference was statistically significant for insoluble fibers (2.30 g/day (IQR 1.48, 2.98) in CD vs. 2.97 g/day (IQR 2.03, 5.24) in HS, *p* = 0.03) but not significant for soluble fibers: 1.13 g/day (IQR 0.82, 1.51) in CD vs. 1.37 g/day (IQR 1.03, 2.12) in HS (*p* = 0.92). As a result, total fiber consumption was lower in CD than in HS, but its statistical significance was borderline (8.25 g/day (IQR 6.42, 9.74) vs. 9.45 g/day (IQR 7.02, 13.01), *p* = 0.056).

### 3.2. Proteins and Amino Acids

Total protein intake was almost significantly different among groups: 58.66 g/day (IQR 53.22, 70.02) in CD vs. 51.95 g/day (IQR 46.04, 59.42) in HS, *p* = 0.056). When considered separately, animal protein consumption was higher in CD than in HS group (29.75 g/day (IQR 24.97, 34.32) vs. 22.92 g/day (IQR 18.62, 28.47), *p* = 0.01), while vegetable proteins were consumed similarly: 11.66 g/day (IQR 8.39, 13.73) vs. 12.1 g/day (IQR 9.05, 6.72) in CD and in HS, respectively. 

Moreover, the ratio between animal and vegetable proteins in the CD group was 2.6, whereas in the HS it was 1.9. When we analyzed the weekly frequencies in the consumption of meat, the CD group showed a similar intake than HS: 10.2 (±2.8) vs. 9.9 (±4.2) times a week, respectively (*p* = 0.839).

We also found different intakes of several amino acids in the two group. Those with a statistically significant difference are listed in Table 1. Leucine (Leu) showed a borderline statistical significance (*p* = 0.055) with a higher consumption in CD than in HS group: 2.61 g/day (IQR 1.99, 1.37) and 2.26 g/day (IQR 1.56, 2.59).

### 3.3. Carbohydrates

No significant differences were observed in carbohydrate consumption between the two groups. However, available carbohydrate intake was higher in CD than in HS: 185.7 g/day (IQR 162.9, 233.5) vs. 165.5 g/day (IQR 150.4, 188.2), respectively (*p* = 0.48). 

We also checked the weekly food frequencies regarding the assumption of sweetened food industry products, including snacks and industrial confectionery. Average oral consumption was of 9.6 (SD ± 4.4) in CD and 2.5 (SD ± 2.8) among the healthy children (*p* < 0.001). 

### 3.4. Lipids

We found a significantly lower consumption of some saturated and unsaturated fatty acids in CD compared to the HS group. In detail, those with a statistically significant difference are listed in Table 2. 

Stearic acid consumption was lower in the CD group than in HS, but the difference was statistically borderline (*p* = 0.059): 0.78 g/day (IQR 0.63, 1.41) and 1.06 g/day (IQR 0.80, 1.64), respectively.

### 3.5. Micronutrients 

Vitamin A intake was significantly lower in CD patients compared to HS: 344.60 mcg/day (IQR 271.81; 353.28) vs. 468.05 microg/day (IQR 271.13, 674.64), *p* = 0.03. 

β-carotene showed a similar trend: 0.26 g/day (IQR 0.06, 2.97) in CD and 0.39 g/day (IQR 0.27, IQR 0.77) in HS, (*p* = 0.04).

On the contrary, Niacin (Vitamin B3) intake was significantly higher in CD children than in HS: 9.84 mg/day (IQR 8.37, 11.71) and 8.04 mg/day (IQR 6.35, 10.49), respectively, *p* = 0.02.

The analysis of polyphenol intake showed lower values in CD than in HS group: 164.22 mg/day (IQR 69.27, 246.11) vs. 289.32 mg/day (IQR 206.85, 404.14), (*p* < 0.005).

We did not observe a significant difference in intakes of the other micronutrients and vitamins between the two groups of subjects.

### 3.6. ORAC and PRAL 

ORAC was significantly lower in CD group than in HS: 1476.5 micromol/day (IQR 860.95, 1951.35) vs. 3128.25 micromol/day (IQR 1944.05, 4009.19) (*p* < 0.01), whereas PRAL was higher in CD group (19.01 pr/day (IQR 14.01, 27.3) vs. 14.08 pr/day (IQR 8.85, 19.82), *p* = 0.003).

## 4. Discussion

Nutrition represents a crucial field of interest in CD. First, because when the disease occurs during childhood and adolescence, nutritional status may have a strong impact on growth and puberty [17]. The combination of inflammation, nutritional inadequacies, reduced food intakes, and the increased energy and nutrient requirements related to age may seriously threaten patients’ healthy growth and development [18]. Second, undernutrition and nutritional deficiencies may result in higher infection rates, prolonged hospitalization and higher rates of postoperative complications [19]. Third, Exclusive Enteral Nutrition (EEN) represents the first treatment option in low-moderate risk pediatric CD patients independently of the disease location to induce remission [20]. It consists of providing the total amount of calories and nutrient requirements of patients through a liquid formula administered orally or via a nasogastric tube [21]. EEN allows to avoid steroid drugs which are associated with growth retardation and bone maturation delay [17,22]. Furthermore, many attempts have been made to reproduce the still largely unknown mechanisms of action of EEN. Such efforts are aimed to overcome barriers such as monotony of EEN and poor palatability experienced by patients, thus providing a nutritional maintenance therapy. Among the several food-based therapies attempted, Crohn’s Disease Exclusion Diet (CDED) currently represents the most clinically documented treatment for induction and maintenance of remission [23,24,25]. CDED-excluded foods are those supposed to facilitate the vicious cycle of the “bacterial penetration hypothesis”. According to this, some intestinal bacterial species might be able to activate the immune system, thus generating inflammation, impairing intestinal permeability, and subsequently increasing the migration of harmful bacteria [26].

In 2018, the Porto IBD group of the European Society of Pediatric Gastroenterology, Hepatology and Nutrition (ESPGHAN) published a position paper on nutrition in IBD that provided recommendations on nutrition surveillance and supplementations [27].

We observed a lower fiber intake in CD, mainly of the insoluble fraction. Fibers are well known to be used by intestinal microorganisms to produce short-chain fatty acids, such as acetate, butyrate, and propionate. Among them, butyrate promotes the integrity of the intestinal barrier by increasing production of mucin, antimicrobial peptides, and the expression of genes coding for proteins involved in tight junctions. Butyrate is also a source of energy for the colonocytes themselves, as well as controlling their proliferation, differentiation, and apoptosis [28]. 

On the other hand, insoluble fiber is well known to cause gastrointestinal symptoms (such as flatus, abdominal tenderness, and bloating) in people suffering from active disease. It has been described that adult CD patients tend to modify their dietary habits to prevent bothersome symptoms [8]. In our experience, this does not happen in children and low consumption of insoluble fibers is associated with the poor consumption of vegetables that is typical of their age and their dietary habits, independently from the disease. 

The high consumption of animal proteins is a typical pattern of the so-called Western diet. 

Epidemiological evidence indicates an association of this dietary pattern with an increased risk of CD, but data are still conflicting, and little is known about the exact mechanisms [7,29]. Two epidemiological studies demonstrated a correlation between the incidence of IBD and high animal protein intake (meat and fish but not eggs and dairy products) [30,31].

A systematic review by Hou and colleagues, collecting 19 studies (only one on children) and 2609 patients with IBD, showed an association between pre-diagnosis dietary habits and the risk of developing IBD. Specifically, they showed an increased risk of CD with high intake of PUFAs, omega-6 fatty acids, saturated fats, and meat. Conversely, a high intake of dietary fibers and fruits was associated with a lower risk of CD [32]. 

In a study by Opstelten et al. the authors observed that, although IBD patients had higher dietary intakes of animal protein than healthy subjects, there was no association between meat consumption and disease relapse [33].

It has been proposed that some amino acid-derived bacterial metabolites might damage the colonic epithelial cells when present in excess [34,35]. The low fiber intake and a high content in animal proteins, saturated fats, and processed and refined foods have been demonstrated to cause a sort of bio-environment selection with a fall in ancient microorganisms, such as fiber-degrading bacteria, and a simultaneous increase of other bacterial groups able to metabolize animal proteins, animal fats, and sugars [36].

However, in a systematic review, Spooren and collaborators concluded that, given the multifactorial etiology of these diseases, no significant association could be found between protein intake and IBD risk [37]. 

Because of the higher animal protein intake, especially meat, several amino acids were also more consumed by our CD patients. 

Nutritional support should be given to families in order to encourage consumption of legumes, preferably hulled or served as soups to make the fiber component more easily digestible. 

In our study, although statistically nonsignificant, we observed that carbohydrate consumption was higher among CD patients as well as sugary snack food products; they are of special interest since they can be considered indirect indicators of the intake of emulsifiers and food additives. 

A systematic review by Hou et al., analyzing the possible association between pre-illness carbohydrate intakes and the risk of developing CD, led to conflicting results [32]. However, among the studies examined, the only one conducted on a pediatric cohort did not find any significant association, even regarding consumption of sweetened sodas, pizzas, hot dogs, and burgers [38].

The dietary intake of fatty acids in CD has been investigated and data are conflicting. A study in adult CD patients reported an association between active disease and a high intake of total, saturated, and mono-unsaturated fats [39]. Associations between CD relapse and fat intake were not found in other studies [33,40]. In the pediatric study conducted by Amre et al., consumption of long-chain omega-3 fatty acids was negatively associated with CD, and a higher ratio omega-3/omega-6 fatty acids was significantly associated with lower risks for CD. The overall intake of fats did not significantly differ and the same applies when they were categorized into monounsaturated and saturated [38]. 

We observed lower Vitamin A and β-carotene intakes in CD patients compared to HS. Vitamin A is a group of organic fat-soluble compounds including retinol, retinoic acid (RA), and several carotenoids including beta-carotene. They mainly come from animal foods. RA is a metabolite of Vitamin A, acting as ligand for RA receptors (RARs) and Retinoic-X Receptor (RXR), transcription factors regulating gene expression. Vitamin A deficiency has been demonstrated to be proinflammatory for the gut due to several immunological mechanisms such as suppression of T regulatory cell activity [41,42]. 

Vitamin A deficiency is frequently found in CD patients [43,44,45]. Both lower serum retinol concentrations and inadequate hepatic vitamin A stores were demonstrated in a case-control study conducted in adults. These data were not associated with dietary intake, disease location, presence of activity, and prior bowel resections [46]. 

We did not observe any difference between CD and HS in terms of Vitamin B12 and folate intakes. Vitamin B12 is mainly available in animal products and is absorbed in the ileum. Folate is available in several vegetables, and it is absorbed in the duodenum and proximal jejunum. Ileal disease or resection may determine vitamin B12 malabsorption, so CD patients may be at risk of deficiency [47]. Vitamin B12 screening should be performed in all patients with ileal or ileocolonic resection and in patients with suspected vitamin B12 malabsorption, and, if depleted, should be administered as a food supplement or by the parenteral route [27,48]. Folate deficiency is frequently detected in IBD patients, especially in CD [49].

However, although serum deficiencies may be detected, oral intakes may still be normal or higher than in healthy subjects. The large consumption of meat in the CD cohort might explain the absence of a difference of vitamin B12 intakes compared to HS. A high intake of Vitamin B12 was also observed in another study conducted on IBD children [12]. 

Niacin was also consumed in greater amounts in CD patients. A study found that two thirds of adult CD patients have low plasma vitamin B3 but no data about oral intake in children with CD are available [44]. 

The analysis of polyphenol intake and ORAC showed lower values in the CD cohort than in the HS group. These organic compounds are a large group of molecules contained in fruits, some vegetables, cocoa, herbs, and spices [50]. Polyphenols seem to be involved in IBD pathogenesis. They are thought to prevent IBD through inhibiting oxidative stress, and the associated damage to macromolecules such as lipids, proteins, and DNA [51]. 

The low consumption of polyphenols is likely reflective of a diet poor in vegetables. Several studies agree with such data [9,38].

Our nutritional software analysis also found a higher PRAL value in CD patients than in HS. This is in line with the high intake of animal proteins we observed in CD children.

In the light of these findings, nutritional assessment and support seem to be crucial in the management of CD patients. Dietary guidance should be provided to families, even if a specific food-based therapy is not followed. The aims of such support are to prevent any macro- and micronutrient deficiencies, frequently re-assess patients’ nutrition status depending on the activity or remission phase of the disease, and early identification of inadequate or imbalanced oral intakes to provide advice and supplementations, accordingly. Early nutritional strategies can lead to a better disease control, as well as catch-up growth, bone mineral density improvement, and adequate pubertal development.

We are aware of several limitations of our study. Firstly, the small number of patients enrolled; secondly, the indirect evaluation of their food intake; and thirdly, the lack of a comparison of our findings with blood concentrations of vitamins and minerals. Moreover, the lack of data concerning healthy subjects’ nutritional status made it impossible to make inferences about their dietary habits.

Because of these limitations, we might have missed some relevant data, such as significant differences in carbohydrates and the presence of a specific dietary lipid profile, as observed elsewhere [34]. 

However, this study’s strength is that although some nutrients could be biochemically measured in the blood, others (such as polyphenols and fibers) could not be easily evaluated, so bromatological analysis indirectly provided useful information.

Finally, we adopted a detailed seven-day food diary to overcome the risk of limited data obtained by a recall questionnaire. 

## 5. Conclusions

Nutritional assessment is crucial in the management of CD patients since a poor nutritional status is associated with growth retardation and disturbance of pubertal development. 

Epidemiological evidence showed an increase of incidence of IBD, including CD, in countries where people have shifted to Western dietary habits, and this represents one of the main clues to the relevant influence that diet has on IBD pathogenesis [21].

Bromatological analysis of a seven-day food diary could be an easy, affordable, and non-invasive tool to provide important information about dietary habits and to assess the risk of macro- and micronutrient deficiencies. Nutritional guidance should always be given to families of CD patients to minimize dietary inadequacies and their consequences. 

Further studies on larger cohorts will be needed to confirm our observations.

## Figures and Tables

**Figure 1 nutrients-14-00499-f001:**
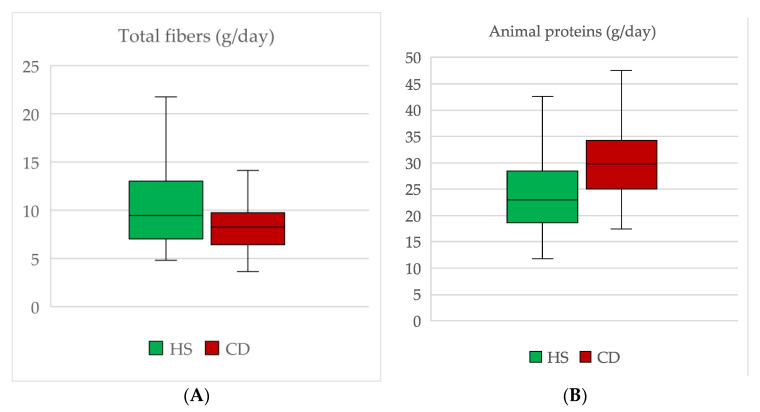
Graphic representation of tbe most relevant statistically significant results. (**A**) Total fibers; (**B**) Animal proteins; (**C**) Vitamin A; (**D**) Polyphenols; (**E**) PRAL; (**F**) ORAC. Boxes represent IQR and horizontal middle lines are medians. Whiskers highlight minimum and maximum values. *p*-values are mentioned in the text. HS: Healthy subjects. CD: Crohn’s disease. ORAC: Oxygen Radical Absorbance Capacity. PRAL: Potential Renal Acid Load.

**Table 1 nutrients-14-00499-t001:** Medians ± IQR of 7-day amino acids oral intake in CD patients and healthy subjects.

	CD (g/day)		HS (g/day)		
	Median	IQR	Median	IQR	*p*
Lys	2.45	1.87, 2.74	1.74	1.25, 2.25	0.004
His	1.10	0.95, 1.27	0.82	0.55, 1.00	0.002
Arg	1.75	1.56, 1.97	1.30	1.06, 1.56	0.001
Asp	2.56	2.12, 2.99	2.11	1.60, 2.53	0.019
Val	1.74	1.51,1.98	0.68	1.07, 1.72	0.015
Gly	1.18	1.12, 1.39	0.99	0.75, 1.14	0.001
Ile	1.54	1.36, 1.72	1.21	0.90, 1.45	0.001
Thr	1.40	1.11, 1.58	1.05	0.81, 1.26	0.002
Ala	1.54	1.26, 1.78	1.12	0.87,1.49	0.002
Met	0.85	0.68, 0.94	0.68	0.46, 0.82	0.015
Tyr	1.26	0.92, 1.37	1.03	0.66, 1.20	0.034

Lys: Lysine; His: Histidine; Arg: Arginine; Asp: Aspartic acid; Val: Valine; Gly: Glycine; Ile: Isoleucine; Thr: Threonine; Ala: Alanine; Met: Methionine; Tyr: Tyrosine; CD: Crohn’s Disease; HS: healthy subjects.

**Table 2 nutrients-14-00499-t002:** Medians ± IQR of 7-day fatty acid oral intake in CD patients and healthy subjects.

	CD (g/day)		HS (g/day)		
	Median	IQR	Median	IQR	*p*
Palmitic	1.94	1.53, 3.23	2.68	2.20, 3.90	0.037
Arachidic	0.02	0.01, 0.02	0.04	0.02, 0.08	0.005
Oleic	5.47	3.21, 7.30	7.45	5.66, 10.93	0.003
Linolenic	0.15	0.09, 0.23	0.21	0.16, 0.26	0.008

## Data Availability

Data will be available at request.

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
