# Peer review of "Dietary Habits of a Group of Children with Crohn’s Disease Compared to Healthy Subjects: Assessment of Risk of Nutritional Deficiencies through a Bromatological Analysis"

_nutrients, 2022, doi:10.3390/nu14030499_

Round 1

Reviewer 1 Report

Comments to the Authors of manuscript number: nutrients-1563741 entitled “Dietary Habits of a Cohort of Children with Crohn’s Disease Compared to Healthy Subjects: the Assessment of Risk of Nutritional Deficiencies through a Bromatological Analysis”.

The authors have presented a study involved children with CD. It is very interesting and worth to publish paper due to the fact that such studies are rare. Authors summarized the whole diet of such children to present many mistakes made by children`s parents, which lead to CD in so young age. It is important finding. Congratulations.

  1. L 18 – it is not clear how many children participated in the study.
  2. L 17 – if examined suffering children from CD was called CD group, this term should be used along the whole text. E.g. not L 18 – 20 patients
  3. L 22 – all abbreviations should be explained when are used for the first time. They are explained L 92 and 93.
  4. L 39 – why Authors introduced suddenly IBD? The study is about CD. There is no explanation.
  5. L 39, 44 – the abbreviation of IBD is not explained.
  6. IBD deferrers from CD
  7. I think that the introduction is unclear due to that Authors did not explain the abbreviation of IBD. They used it as inflammatory bowel disease, but is can be linked with irritated bowel disease or syndrome.

Author Response

Dear Reviewer,

thank you for your suggestions. 

  1. Sixty-eight children partecipated in the study.
  2.  At the beginning of the revised introduction, I better defined CD as belonging to Inflammatory Bowel Disease (IBD). Diet contribution to IBD pathogenesis is thought to be similar in both IBD (Crohn and Ulcerative Colitis) and both conditions are studied by papers that we mentioned in our discussion.
  3. Abbreviations were clarified. In our paper, IBD refers to Inflammatory Bowel Disease. 

Thank you!

Best Regards,

Flavio Labriola

Reviewer 2 Report

In the manuscript, Flavio Labriola et al. describe a cross-sectional analysis using data from a single center.

The authors compared nutrient intakes between 20 CD patients and 48 healthy individuals.

Since this is a cross-sectional study and not a case-control study, the authors should avoid using the term “healthy controls” and instead use “healthy individuals” or “healthy subjects”.

Overall, the manuscript is well written. However, several major revisions are needed.

There are large cohorts of patients with CD who are under study. I think calling 20 patients a “cohort”
 is an overestimation of the study and the authors should change accordingly (maybe “group” is more suitable).

The authors mentioned “matched” healthy controls in the end of the introduction section and in the first sentence of the result section (lines 53+108). The statistical test performed is not suitable for paired samples. The authors should clarify this point in the method section and describe how the matching was performed (age – sex matched?). In addition, they should reanalyze the data and update the manuscript accordingly (Wilcoxon test will be appropriate).

In the method section, the authors show add sample size calculation, especially since this study may lack power to detect small differences.

The result section lacks p-value (lines 116-117), the authors cannot just state “showed no significant differences” since p-value of 0.06 is not the same as 0.6 (although, both are not statistically significant). I believe that “borderline” significance (p-values between 0.05 and 0.1) is very important in this study due to the limited sample size.

An average should always be presented with SD (lines 113-114, PCDAI).

The authors stated they are aware of the several limitations (lines 319-323) and they should explain if and how these limitations may influence the study results.

Author Response

Dear Reviewer,

Thank you for Your suggestions.

We recognize that the terms "cohort" and "control" might be inappropriate for our study, because of the small samples. We replaced them with the terms "group" and "healthy subjects".

In our study, CD children and healthy subjects have different mean age, although they all belong to the pediatric age. Instead, sex distribution do not significantly differ between the two groups. Anyway, we admit that the term matched might be confusing so we delated it. 

Furthermore, since the samples are independent (i.e. unpaired) and distributions were not normal, we chose Mann-Whitney test instead of Wilcoxon test. 

We are aware of limitations regarding the small sample size and probably we might have missed some interesting data due to non-statistically significant results such as significant differences in carbohydrate and the presence of a specific dietary lipid profile. We added more detailed conclusions.

Standard deviations were added to average PCDAI. 

Thank you 

Best regards,

Flavio Labriola